# Parental Phasing Study Identified Lineage-Specific Variants Associated with Gene Expression and Epigenetic Modifications in European–Chinese Hybrid Pigs

**DOI:** 10.3390/ani15101494

**Published:** 2025-05-21

**Authors:** Chenyu Li, Mei Ge, Keren Long, Ziyin Han, Jing Li, Mingzhou Li, Zhiyan Zhang

**Affiliations:** 1National Key Laboratory for Swine Genetic Improvement and Germplasm innovation Technology, Jiangxi Agricultural University, Nanchang 330045, China; 18702523671li@gmail.com (C.L.); ·; 2State Key Laboratory of Swine and Poultry Breeding Industry, College of Animal Science and Technology, Sichuan Agricultural University, Chengdu 611130, China; keren.long@sicau.edu.cn (K.L.); ziyinhan@126.com (Z.H.); lijing_2020@sicau.edu.cn (J.L.)

**Keywords:** lineage-specific genetic variants, phased gene expression, phased epigenetic regulation, hybrid pigs

## Abstract

Crossbreeding pigs from different breeds can produce offspring with economic traits comparable to or even better than those of their parents. However, the genetic and epigenetic mechanisms underlying these advantages and their impact on gene expression remain to be fully elucidated. Therefore, we selected hybrids pigs born from crosses between European and Chinese lineages and conducted phasing analysis to trace the parental origin of genetic variants—including single nucleotide polymorphisms (SNPs) and structural variations (SVs)—as well as gene expression and epigenetic modifications. Notably, we found that genetic variants inherited from different lineages were significantly associated with phased gene expression and histone modification patterns. By integrating these association results, we inferred potential regulatory mechanisms by the lineage-specific genetic variants and that epigenetic modifications influence lineage-specific gene expression. These findings were also further supported by results from larger population studies. This work elucidated the profound impact of inherited genetic variation on gene regulatory mechanisms in hybrid pigs, offering a theoretical foundation and practical guidance for advancing pig breeding through molecular selection.

## 1. Introduction

Genetic variation between parental phases in hybrid individuals can drive differences in gene expression by regulating cis-regulatory elements such as enhancers, promoters, and insulators [1]. The accurate identification of genetic variants and high-efficiency phasing of hybrid genomes are critical for deciphering the mechanisms underlying parentally biased gene expression [2]. This understanding not only elucidates genetic regulatory patterns across different breeds but also provides deeper insights into epigenetic regulation during livestock domestication.

Single nucleotide polymorphisms (SNPs) are the most common type of genetic variation in the genome, playing a key role in deciphering heterosis and population genetic structure. As an important indicator of genomic heterozygosity, SNPs are widely used to assess the genetic diversity of hybrid individuals [3]. Structural variants (SVs) represent another important class of genetic variation that plays a crucial role in trait regulation. It also closely associated with heterosis and economically important traits [4]. SVs, including large-scale insertions, deletions, inversions, and duplications, can not only directly affect gene coding functions but also modify regulatory elements, thereby indirectly influencing gene expression [5]. Recent advances in third-generation sequencing technologies, such as PacBio and Oxford Nanopore sequencing, have significantly improved the sensitivity and accuracy of SV detection, enabling researchers to precisely identify large insertions and deletions that are difficult to detect using conventional short-read sequencing [6]. Moreover, these technologies provide higher resolution in repetitive and low-complexity regions, which are often enriched for SVs [7]. Compared to SNPs, SVs offer unique advantages in genetic studies, for example, SVs exert broader effects, as they can directly alter gene dosage or regulatory element functions, making them highly relevant to dissecting the heritability of complex traits [8,9].

Lineage-specific genetic variants are particularly relevant to the study of heterosis in pigs. Currently, commercial crossbred pigs are typically derived from parental populations with distinct genetic backgrounds. Differences in genetic variants between parental phases may contribute to the unique trait advantages of hybrid offspring through various mechanisms. For instance, several representative Chinese indigenous pig breeds, such as Erhualian and Meishan, are well known for their high reproductive performance and enhanced fat deposition, whereas European commercial pigs, including breeds like Landrace and Duroc, are characterized by faster growth rates and higher lean meat yield [10,11,12]. Lineage-specific genetic variants can integrate beneficial traits from both parents to optimize regulatory networks, thereby enhancing adaptability and growth performance in hybrid pigs. Moreover, lineage-specific genetic variants may potentiate the expression of trait-associated genes or remodel regulatory elements, thereby enhancing growth, metabolic efficiency, and behavioral adaptability—lineage-specific advantages that offer a unique perspective on the molecular basis of heterosis [13].

In agricultural breeding programs for plants and animals, the superior traits of hybrid offspring are often associated with gene expression differences between paternal and maternal lineages. Functional annotation and mechanistic analysis of regulatory elements at the haplotype level allow for more precise identification of parental alleles that contribute to specific traits, such as disease resistance, productivity, or product quality. Li et al. conducted a comprehensive haplotype-resolved annotation of H3K27me3 in the hybrid rice line [14]. In studies of major livestock (pigs [15]) and poultry (chickens [16]), researchers similarly used hybrid offspring to distinguish between parental genomic contributions at the haplotype level. They examined multiple histone modifications, primarily H3K4me3 and H3K27ac, and observed signal differences between paternal and maternal haplotypes. These findings underscore the importance of integrating diverse histone modification types to characterize haplotype-specific regulatory mechanisms.

In this study, we established three kinds of European–Chinese crossbred families for phasing analysis of genomic, transcriptomic, and epigenomic data. By leveraging phased genetic variants, we integrated colocalization analyses with phased gene expression and epigenetic modifications to investigate how lineage-specific variants influence gene expression and to explore the underlying regulatory mechanisms.

## 2. Materials and Methods

### 2.1. Biological Sample Collection and Family Identification

In this study, three hybrid families were generated by crossing representative European commercial breeds with Chinese indigenous breeds. The paternal lines included Duroc Large White and Berkshire, which are commonly used in commercial pig breeding due to their well established performance in growth and carcass traits. The maternal lines consisted of three geographically distinct Chinese native breeds. Erhualian pigs, renowned for their high reproductive capacity, originate from Jiangsu Province in eastern China. GanXi pigs, characterized by their typical “two-end black” appearance, are native to western Jiangxi Province. LiangGuang pigs, distinguished by their small ears and piebald coat, are distributed across Guangdong and Guangxi Provinces in southern China. We collected semen from the father, ear tissue from the mother, and backfat (BF) and longissimus dorsi (LD) muscle samples from hybrid offspring at 90–100 days in the trio family of each crossbreeding lines. The samples were snap-frozen in liquid nitrogen and stored at −80 °C. To verify true hybrids and family information, we estimated breed proportion and parentage assignment in each family. SNPs were genotyped using CC1 Porcine SNP50K BeadChip (Illumina, San Diego, CA, USA), and Genome Studio software (v1.0) was employed to call the genotypes of SNPs [17]. Breed proportion was estimated by using the ADMIXTURE (V1.3.0) [18] program with default parameter and IBD; Mendelian errors were calculated with Plink (V1.9) [19].

### 2.2. SNP Calling and Phased SNP Calling

A total of 16 pigs, including both parents and offspring, were used for short-read whole-genome sequencing (WGS), while long-read WGS was performed on 6 offspring individuals. Genomic DNA from all offspring was extracted from longissimus dorsi (LD) muscle. Whole-genome short-read libraries were sequenced on the Illumina HiSeq 4000 platform (Illumina, San Diego, CA, USA) (PE150), and long-read sequencing was performed on the ONT PromethION platform. WGS data mapping was performed using Fastp v0.23.2 to trim adapters and filter out reads shorter than 100 bp (-l 100) from short-read WGS (Illumina) data of each family. High-quality reads were then mapped to the susScr11 reference genome using BWA v0.7.17 mem [20], and reads with a mapping quality score (MAPQ > 30) were extracted. SAMTools v1.9 [21] was used to filter PCR duplicates and sort and convert the data into BAM files. For the PromethION long-read sequencing data of the hybrid offspring, reads were mapped to susScr11 using Minimap2 v2.17 [22] with the “-ax map-ont” option, followed by merging and sorting with SAMTools [21]. SNP calling was performed using GATK v4.1.4.1 [23], where AddOrReplaceReadGroups and MarkDuplicates were used to add sample group information and mark PCR duplicates. SNPs were identified using GATK HaplotypeCaller and SelectVariants, and high-quality heterozygous SNPs were retained for subsequent phasing analysis. For SNP phasing, we prepared the variants.vcf file (a VCF-format variant file recording genetic variations in the trio family), father.bam, mother.bam, offspring.bam1 (short-read WGS), and offspring.bam2 (long-read WGS) mapping files, as well as the trio.ped file (describing the kinship relationships among the samples) and ref.fasta (the reference genome in FASTA format). We performed phasing on the diploid offspring sample using WhatsHap phase v1.4 [24] with the following command to generate the phased SNPs: “whatshap phase --reference=ref.fasta --ped=trio.ped -o phased.vcf variants.vcf father.bam mother.bam offspring.bam”.

### 2.3. SV Calling and Phased SV Calling

Sniffles2 V2.2 [25] was executed with default parameters on the BAM files generated by Minimap2 to identify structural variants (SVs) ≥ 50 bp in size for each sample, including deletions, insertions, inversions, duplications, and translocations. For phased SV calling, the mapped BAM files were phased into paternal.bam and maternal.bam using the Whatshap v1.4 [24] haplotag tool, incorporating phased SNP information. Similarly, the phased BAM files were conducted to single-sample SV calling using Sniffles2 V2.2. Subsequently, multiple phased sample.snf files were processed using Sniffles2 V2.2 for combined SV calling. Finally, SVs with a variant frequency greater than 10% were retained and processed through filtering to generate a variant matrix.

### 2.4. Genome Transposon Annotation

The known transposable elements (TEs) in the genome were annotated using RepeatMasker v4.1.5 (https://www.repeatmasker.org/, accessed on 10 March 2025) in combination with the Repbase database [26]. Default parameters were used to classify transposons into different families and superfamilies.

### 2.5. Quantification of Phased Gene Expression

RNA-seq was performed on six hybrid offspring, with both backfat (BF) and longissimus dorsi (LD) muscle collected from each individual, resulting in a total of twelve samples. Total RNA was extracted and purified; mRNA was enriched using oligo(dT) beads and converted into cDNA. Libraries were constructed using the NEBNext^®^ Ultra™ RNA Library Prep Kit for Illumina (New England Biolabs, Ipswich, MA, USA). RNA-seq libraries were sequenced on the Illumina HiSeq 4000 platform using 150 bp paired-end reads. The raw RNA-seq data were initially filtered using Fastp v0.23.2 [27] with the parameter option “-l 100”. The filtered sequencing files were then mapped to the pig reference genome using HISAT2 v2.2.0 [28] and SAMTools v1.9 [21] was used to sort and convert the data into BAM files. Similarly to the phasing method used for ONT data, the BAM files were phased into paternal.bam and maternal.bam using Whatshap v1.4 [24] and heterozygous SNP information from the hybrid offspring. Subsequently, each phased BAM file was quantified using StringTie v2.2.1 [29]. Genes with a read count of fewer than 10 in either phase were filtered out.

### 2.6. Chromatin State Prediction and Identification of Phased Histone Modifications and CTCF Binding

Cut & Tag was conducted on five hybrid offspring, with a total of 50 samples from backfat (BF) and longissimus dorsi (LD) muscle, to profile multiple histone modifications and CTCF binding. Libraries were constructed for each target and sequenced on the Illumina NovaSeq 6000 platform (Illumina, San Diego, CA, USA) using 150 bp paired-end reads. The raw sequencing data for each assay were initially filtered using Fastp v0.23.2 [27] with the optional parameter “-l 100”. The filtered sequencing files were then mapped to the pig reference genome using BWA and SAMTools v1.9 [21] was used to filter PCR duplicates and sort and convert the data into BAM files. MACS2 v2.1.1 [30] was used for peak calling with the optional parameter “f BAMPE -q 0.05 --nomodel --shift -0 --keep-dup all”. Next, we used DiffBind v3.2.0 [31] to obtain consensus peaks for each assay across different samples within the same tissue, as well as consensus peaks across all assays. Based on these consensus peaks, we applied ChromHMM [32] to predict chromatin states across the genome using 500 bp windows for the corresponding tissue. Additionally, the phasing of Cut & Tag data was performed similarly to ONT sequencing. Each assay’s mapped BAM file was phased into paternal.bam and maternal.bam using Whatshap v1.4 [24] and phased SNP information. MACS2 v2.1.1 [30] was then used with the same parameter as before to perform peak calling on the phased BAM files. Finally, each assay’s consensus peak was used to filter the corresponding phased peaks, yielding the filtered phased peaks.

### 2.7. Hi-C Data Analysis and Chromatic 3D Structure Identification

We isolated cells from the backfat tissue of an adult boar and performed an in situ Hi-C experiment to analyze its three-dimensional genome architecture. The experimental procedure was optimized based on the method described by Rao et al. [33]. The sequencing library was constructed using the NEBNext Ultra DNA Library Prep Kit (New England Biolabs, Ipswich, MA, USA) and subjected to paired-end 150 bp (PE150) sequencing. Raw sequencing data were initially processed using Fastp v0.23.2 [27] with the optional parameter “-l 100” for preliminary read filtering. Subsequently, Juicer v2.0 [34] was used to process the sequencing data for each library, including adapter trimming, low-quality read filtering, mapping (BWA-MEM to the pig reference genome Sus scrofa 11.1), and restriction site identification (HindIII). The final interaction file (“merge30.txt”) was generated with a mapping quality threshold of MAPQ > 30. After merging interaction files from individual libraries, the calculate_map_resolution.sh script in Juicer was used to estimate the maximum resolution, which was approximately 4.5 kb. Finally, Juicer Tools v1.22.01 was used to generate interaction matrices at resolutions of 25 kb and 5 kb. The topologically associated domains (TADs) were called with HiCexplorer v3.7.2 [35] hicFindTADs with default parameters at a resolution of 25 kb. Loops were identified by Mustache v1.0.1 [36] with resolution parameter “--resolution 5000” and others optional parameters “-p Threshold 0.05 -normalization weight”.

### 2.8. Colocalization of Phased Genetic Variants with Phased Gene Expression and Phased Epigenetic Modifications

We conducted eQTL and epiQTL analyses using the colocalization analysis method of MatrixeQTL v2.3 [37] (R 4.1.1). Gene expression levels for each phase were quantified in TPM (Transcripts Per Million) to construct an expression matrix, which was then log-transformed for normalization. For histone modifications and CTCF signals, using bedtools v2.25.0 coverage [38], we calculated the reads coverage of each phased BAM file within the corresponding phased peak. The normalized reads coverage was also quantified in TPM. We then performed linear regression analyses in MatrixeQTL v2.3 [37], associating phased genetic variants with phased gene expression levels, as well as phased histone modifications and CTCF signal intensities. Sex and batch were considered as covariates and included in the model to correct for potential confounders. Significant associations were defined as those with an FDR less than 0.05, a statistical method that controls for multiple testing by limiting the expected proportion of false positives [39].

### 2.9. Prediction of Transcription Factor Motifs

Transcription factor motif prediction was performed using the MEME Suite (https://meme-suite.org, accessed on 10 March 2025) online tool. First, the target DNA sequences in peaks were formatted into FASTA using SeqKit v2.9.0 [40]. The FASTA file was then uploaded to the MEME motif discovery tool, with the analysis mode set to Classic mode. The identified motifs were subsequently matched against known motifs in the JASPAR database using the TOMTOM tool in MEME Suite. The first three positions of non-repetitive motifs were considered potential biological functional sites and associated transcription factors.

### 2.10. Functional Annotation of Single-Cell Data

The single-cell data and cell type annotations for human adipocyte nuclei and skeletal muscle were obtained from publicly available datasets CellxGene (https://cellxgene.cziscience.com/, accessed on 10 March 2025) and Disco (https://www.immunesinglecell.org/, accessed on 10 March 2025), respectively. UMAP was used for clustering different cell types [41], while cell type-specific gene enrichment was visualized using the FeaturePlot function in the Seurat v5.2.1 R package [42].

## 3. Results

### 3.1. Experimental Design and Identification of Genetic Variants Form Different Lineages

To systematically dissect the genetic contributions of European commercial and Chinese indigenous pig lineages in hybrid pigs, we designed three distinct crossbreeding lines: Large White × Erhualian, Berkshire × Ganxi, and Duroc × Liangguang. Each cross included six trio families, with offspring slaughtered at 90–100 days. Samples from backfat (BF) and longissimus dorsi (LD) muscle were collected, accompanied by carcass trait measurements (Appendix A). Individuals were further verified for parental–offspring relatedness using genetic markers; two trio families of each cross were selected for further analysis (Figure 1a). Subsequently, short-read whole-genome sequencing (WGS) was performed for parents, while offspring underwent both short-read and long-read (ONT) WGS. RNA sequencing was conducted for backfat (BF) and longissimus dorsi (LD) muscle, along with Cut&Tag sequencing targeting four histone modifications (H3K4me3, H3K27ac, H3K4me1, and H3K27me3) and one transcription factor (CTCF). To achieve phase-resolved insights into gene expression and epigenetic modifications, we initially conducted genome phasing and identified phased genetic variants. SNP calling was performed for parents and their offspring in each trio family. Additionally, long-read sequencing (ONT) from the offspring was used to generate local genomic segments, thereby enhancing the phasing efficiency of SNPs (Figure 1b). On average, 11,084,942 phased SNPs were identified for each hybrid offspring, and these SNPs exhibited clear clustering according to the phase from different lineages (Appendix A). Leveraging the phased SNPs and ONT data, SVs were also phased (Figure 1b) [43], reaching an average phasing efficiency of 73.43% (Figure 1c), comparable with previous reports [44]. In total, we identified 43,410 and 60,658 deletions (DELs), 45,502 and 66,845 insertions (INSs), 427 and 703 inversions (INVs), and 298 and 404 duplications (DUPs) in the European commercial (paternal) and Chinese indigenous (maternal) lineages, respectively. The increased detection of SVs in the Chinese indigenous lineage may result from both the use of a European commercial pig reference genome and the inherently greater genetic complexity of Chinese indigenous pigs. The length distributions of SVs from different lineages exhibited similar patterns, predominantly ranging from 50 to 2000 bp, with enrichment observed at 50–200 bp and 250–400 bp (Figure 1d). The phased SVs also exhibited well defined clustering patterns according to phases from different lineages, with more distinct clustering observed among Chinese indigenous lineages compared to European commercial lineages (Figure 1e). To further explore lineage specificity in genetic variation, we identified SNPs and SVs that were exclusively present within the same lineage. While lineage-specific SNPs did not exhibit a distinct classification pattern, the number of lineage-specific SVs were generally higher in the Chinese indigenous lineage compared to the European lineage (Figure 1f, Appendix A). It may suggest that, compared to SNPs, large-scale SVs may play a more significant role in driving population differentiation within lineages [45]. An analysis of the genomic distribution of lineage-specific variants revealed that they were primarily located in intronic and intergenic regions, with lower enrichment in exon (Figure 1f, Appendix A). This trend is consistent with observations in human genome studies [46]. Given the critical role of transposable elements (TEs) in genome rearrangement and formation of structural variation, we further examined the distribution of lineage-specific SVs across TE subtypes. Our analysis showed that, on average, 64.53% of lineage-specific SVs were located within TEs, with LINE and SINE elements emerging as major contributors to structural variation. This underscores the role of transposable elements in genome dynamics [47]. Additionally, to demonstrate the ability to precisely detect phased large-range SVs, we separately examined maternal-specific and LG lineage-specific loss of heterozygosity (LOH), both exceeding 1 kb in length (Figure 1g), which were accurately identified. The systematic use of distant hybrid families enabled the comprehensive identification of genetic variations from different European and Chinese lineages, offering critical insights into the genetic basis and functional consequences of hybridization.

### 3.2. Colocalization of Phased Genetic Variants and Phased Gene Expression

To further investigate the relationship between genetic variations and gene expression from different lineages, we performed phased expression analysis in hybrid offspring. On average, 10,797 protein-coding genes were phased and quantified separately for each phase, accounting for 55.86% of autosomal genes. Subsequently, we conducted phase-level eQTL colocalization analysis by integrating phased SNPs, SVs, and phased gene expression. Within a 1 Mb window upstream and downstream of genes, we identified phased SNPs significantly associated with gene expression in BF (n = 7939) and LD (n = 10,644) (FDR < 0.05) (see Section 2), as well as phased SVs in BF (n = 64) and LD (n = 23) (FDR < 0.05) (Appendix A). Notably, colocalizations between phased genetic variants and phased gene expressions were lineage-specific, including *AMIGO2* (adhesion molecule with Ig like domain 2), *KIT* (KIT proto-onco, receptor tyrosine kinase), *OR6N2* (olfactory receptor family 6 subfamily N member 2), *FLVCR2* (FLVCR choline and putative heme transporter 2), *CLU* (clusterin), *PCED1B* (PC-esterase domain containing 1B), *CAB39L* (calcium binding protein 39 like) and *RP9* (RP9 pre-mRNA splicing factor) in BF, and *TAF1D* (TATA-box binding protein associated factor) and *KIT* in LD (Figure 2a). Additional functional annotation was performed by integrating single-cell transcriptomic data from human adipocytes [48] and skeletal muscle [49] along with GWAS datasets (https://pigbiobank.ipiginc.com/home, https://www.ebi.ac.uk/gwas/home, accessed on 10 March 2025). In adipocytes, *AMIGO2* was predominantly expressed in subcutaneous adipocyte (Figure 2c) and was significantly associated with the muscle-to-fat ratio (−log10P = 3.177) in pheGWAS data and BMI (−log10P = 9.3977) in human GWAS datasets. *CLU*, specifically expressed in endothelial cell and mesenchymal stem cell of adipose tissue (Figure 2d), has been reported to promote preadipocyte differentiation when overexpressed, while its knockdown suppresses adipogenesis [50]. These findings suggest that *AMIGO2* and *CLU* may be involved in metabolism and early differentiation of adipocytes, thereby regulating important traits such as development and body size. Furthermore, *KIT*, consistent with previous reports, was highly enriched in mast cells, suggesting its involvement in adipose tissue vascularization and extracellular matrix remodeling, which may affect local tissue architecture and function (Appendix A) [51]. *PCED1B*, *FLVCR2*, and *RP9* are not only specifically expressed in mesenchymal stem cells of adipose tissue but are also expressed in memory T cells, B cells, and other immune cells (Appendix A). This suggests that they may play roles in differentiation and immune regulation within adipose tissue. In skeletal muscle, *KIT* exhibited low expression across multiple cell types (Appendix A), yet previous studies have reported that the *SCF*/*KIT* signaling pathway regulates mitochondrial function and energy expenditure, and is associated with reduced PGC-1α expression and mitochondrial dysfunction in muscle [52]. In contrast, *TAF1D* was highly enriched in multiple cell types (Appendix A). These findings indicate that regulatory regions may harbor potential eQTLs, influencing gene expression and metabolic traits in adipose and muscle tissue. However, the potential involvement of these eQTLs in cis-regulatory elements, chromatin interactions, and their specific functional roles remains to be further investigated in order to elucidate how genetic variants shape the epigenetic landscape through regulatory networks.

### 3.3. Associations Between Phased Genetic Variants and Chromatin State

To investigate the distribution of genetic variants across different chromatin states and their potential impact on regulatory elements, we performed chromatin state prediction in hybrid offspring based on four histone modifications (H3K4me3, H3K27ac, H3K4me1, and H3K27me3) and one transcription factor (CTCF). The whole genome was segmented into 500 bp bins, resulting in the classification of 16 chromatin states, including Active Enhancers, active promoters, CTCF/Active TSS, CTCF/Enhancer, CTCF/Primed Enhancer, CTCF/Promoter, CTCF/Weak Polycomb repression, enhancers, Flanking TSS, insulator, Low Signal, Poised promoters, Primed Enhancers, promoters, Repressed Polycomb, and Quiescent states. We then analyzed the chromatin states associated with genetic variants. The results revealed that, in both BF and LD tissues, SNPs were predominantly enriched in CTCF/Enhancer, CTCF/Active TSS, and Repressed Polycomb (Figure 3a). In contrast, SVs were primarily distributed in Quiescent states, Low Signal, and Weak Repressed Polycomb, accounting for an obviously higher proportion compared to other states (Figure 3b). Compared to SNPs, SVs were less frequently found in active promoters and enhancers, which is consistent with previous report in a large population [45]. It also implies that SNPs tend to occur within regions marked by epigenetic signals, while SVs may be associated with them.

To further discover the associations between genetic variants and histone modifications and CTCF binding from different lineages, we performed phasing analyses of histone modifications and CTCF signals. In BF and LD tissues, we identified an average of 27,412 phased H3K4me3, 75,797 phased H3K27ac, 92,087 phased H3K4me1, 48,416 phased H3K27me3, and 39,990 phased CTCF peaks. We then performed co-localization analysis using phased genomic variants (SNPs and SVs) with phased histone modifications and CTCF peaks, respectively. For SNPs, we directly examined whether they overlapped with these peaks. For SVs, since most are located in inactive chromatin regions, we checked whether phased SVs were within ±1 Mb of these peaks. We identified phased SNPs significantly associated with phased epigenetic modifications (SNP-epiQTLs), including H3K4me3 (BF: 145, LD: 181), H3K27ac (BF: 51, LD: 301), H3K4me1 (BF: 586, LD: 768), H3K27me3 (BF: 147, LD: 160), and CTCF binding (BF: 80, LD: 59) (FDR < 0.05; Figure 3c; Appendix A). Similarly, phased SVs were significantly associated with the phased assays (SV-epiQTLs), with counts of H3K4me3 (BF: 551, LD: 734), H3K27ac (BF: 1,715, LD: 1,513), H3K4me1 (BF: 314, LD: 644), H3K27me3 (BF: 48, LD: 235), and CTCF (BF: 3, LD: 417) (FDR < 0.05; Figure 3d; Appendix A). Interestingly, the phased SVs were also significantly associated with active chromatin regions, particularly those marked by H3K4me3 and H3K27ac, suggesting that SVs may have the potential to serve as molecular markers involved in active regulatory processes (Figure 3d). These findings indicate that phased variants and epigenetic modifications from different lineages are associated, and genetic variants may help us better understand the formation of lineage-specific gene expression by altering the activity of regulatory elements or serving as molecular markers on lineage-specific phase.

### 3.4. Lineage-Specific Genetic Variants Modulate Epigenetic Modification to Regulate Lineage-Specific Gene Expression

Histone modification and transcription factor differences between phases can lead to variations in regulatory elements, resulting in phase imbalanced gene expression and ultimately influencing lineage-specific gene expression [14]. To further investigate how phased genetic variants modulate gene expression through epigenetic modifications and assess the potential as molecular markers, we performed variant screening by integrating eQTLs and epiQTLs. We identified multiple genes, including *AMIGO2*, *KIT*, *ENSSSCG00000036983*, and *PSME3*, whose eQTL regions overlapped with the epiQTL regions (Table 1). In particular, *KIT* exhibited LW lineage-specific high expression in both BF and LD, with a LW lineage-specific duplication (*DUP*, chr8:41,223,208–41,783,661) detected within the *KIT* locus. This finding aligns with previous studies implicating *KIT* in coat color variation, where this SV differentiates the LW lineage from darker-skin pig breeds [53,54]. Further analysis revealed that within the duplicated region, the promoter of *KIT* in the LW lineage exhibited stronger H3K4me3 and H3K27ac signals (Figure 4a). It suggests that, in addition to copy number expansion, lineage-specific promoter activity may also be a key factor contributing to the high expression of *KIT* in the LW lineage. Furthermore, in BF, *AMIGO2*, which exhibits DRC lineage-specific high expression, was found to be flanked (within 1 Mb) by a DRC-specific insertion (*INS*, *chr5:77,568,931–77,569,226*) and SNPs (*a: chr5:77,754,641–77,754,642* and *b: chr5:77,757,817–77,757,818*) (Figure 4b,c). These genetic variants were significantly associated with two DRC-specific H3K27ac signals upstream of *AMIGO2*. As the SNPs were located within H3K27ac peaks, we performed motif prediction for peaks covering *a* and *b* and found that the SNP *a* was located in the core motif of the transcription factor *CEBPB*, while the SNP *b* was located within the core motif for *KLF4* (Figure 4d). Moreover, Hi-C data from pig adipose tissue showed that the H3K27ac peaks covering *a* and *b* are located within the same TAD as *AMIGO2*. A chromatin loop was also identified connecting the H3K27ac peak covering *b* and the promoter of *AMIGO2*, indicating that the three-dimensional chromatin structure provides a spatial basis for the cis-regulation of *AMIGO2*. We also analyzed allele frequencies using previously published whole-genome resequencing data from diverse pig breeds (Table 2) [55,56,57,58]. The results indicated that the Duroc population had a Ref allele frequency exceeding 90.00% for the SNPs and 71.08% for the SV, whereas in other populations, the frequencies were below 30% for both variants. This trend aligned well with our observations. Additionally, by querying the pigGTEx database [59], we identified multiple *AMIGO2* eQTLs (*chr5:77754898-77754899*, *chr5:77754675-77754675* and *chr5:77754686-77754687*) in adipose tissue, which are located within the H3K27ac peak covering *a.* These results suggest that DRC lineage-specific expression of *AMIGO2* may be driven by genetic variants that modulate regulation of enhancer marked by H3K27ac, providing the potential transcription factor binding sites, chromatin interactions, and molecular marker candidates. These findings offer new insights into the genetic regulatory mechanisms of *AMIGO2* and provide potential molecular markers for lineage-specific trait selection in Duroc pigs.

## 4. Discussion

In this study, we systematically analyzed phased genetic variations in European–Chinese hybrid pigs, characterizing their distribution as different distinct lineages and assessing their associations with gene expression and epigenetic modifications. Our findings not only reveal differences in genetic variations from different lineages but also provide empirical evidence for the role of it in modulating gene expression and chromatin organization. These insights advance our understanding of the complexity of the genetic background in gene regulation and offer potential molecular markers for breeding programs.

The effects of parentally inherited genetic variants on offspring phenotypes have long been a research focus. Accurate phasing of genetic variants is crucial for resolving their parental origins. While short-read whole-genome sequencing (WGS) in trio-based family designs allows partial phasing of SNPs, the accuracy of SV identification using short-reads sequencing is limited for variants exceeding read length. Recent advances in long-read sequencing have facilitated the phasing of phase-specific SVs in humans, primarily through computational haplotype block construction [60,61]. However, these studies often lack family-based phasing information. In contrast, family pedigree data are more accessible in animal studies, as demonstrated in cattle, where structural variants have been phased based on hybrid offspring with distinct parental origins [62]. Therefore, we integrated the high sequencing depth of second-generation sequencing with the long-range phasing capability of third-generation sequencing, while incorporating pedigree information to trace the origin of genetic variations in European–Chinese hybrid pigs. This approach provides a deeper understanding of how phased genetic variations influence gene expression and epigenetic modifications, offering valuable genomic resources for germplasm integration and breeding programs.

Compared with conventional methods [63,64,65], eQTL mapping using phased gene expression offers significant advantages, primarily by greatly enhancing statistical power [66]. Concerning the assignment of parental phases, it not only enhances detection sensitivity but also enables the precise dissection of lineage-specific regulatory mechanisms. Moreover, phasing analysis provides insights into how phased epigenetic modifications influence gene expression, thereby establishing a comprehensive regulatory cascade from SNPs to histone modifications and ultimately to gene expression. In the context of hybrid studies, elucidating lineage-specific regulatory patterns derived from distinct genetic lineages is crucial for understanding how hybrid individuals integrate parental advantages to optimize key economic traits.

In this study, we identified lineage-specific expression patterns of *KIT* and *AMIGO2* genes and explored their potential regulatory mechanisms, providing novel insights into how genetic variations and chromatin modifications influence lineage-specific gene expression.

In the Large White lineage, large segmental duplications in the *KIT* gene region may lead to significantly enhanced chromatin activity in its promoter region, making it more active compared to other lineages. This corresponds with the high lineage-specific expression of *KIT* in both BF and LD tissues of the Large White pigs. Studies have shown that abnormal *KIT* expression in mouse adipose tissue can impair the development and deposition of fat tissue [67]. Additionally, the loss of *KIT* expression in human and mouse cardiac muscle tissues has been reported to cause insufficient proliferation of satellite cells, thereby affecting muscle repair and regeneration [63,64]. For immune function, single-cell data from humans show that *KIT* is highly enriched in mast cells, and our previous GWAS findings show that *KIT* is a candidate gene associated with granulocyte count percentage (GRAR) [68]. Importantly, KIT has also been implicated in muscle satellite cell function and adipose tissue remodeling in large-animal models. For instance, Ma et al. reported that *KIT* is the key genes of subcutaneous fat in cattle [69], suggesting a conserved role in livestock fat deposition. In pigs, CNVs and duplications in the *KIT* locus have been associated not only with coat color variation but also with differences in body composition traits, including backfat thickness and carcass quality [70]. Recent analyses further suggest that *KIT* may regulate early adipocyte differentiation through crosstalk with the Wnt and PI3K-Akt pathways, which are key to fat tissue development [71,72]. These evidences suggest that beyond its well-known role in coat color formation [54,73], *KIT* may also play a critical biological role in fat deposition, muscle cell differentiation, and immune function in pigs. Moreover, we observed colocalization of *AMIGO2* gene, specifically highly expressed in the DRC lineage, with lineage-specific genetic variants and epigenetic modifications. Although research on *AMIGO2* in agricultural animals remains limited, it is considered to be highly associated with cellular development and metabolism in medical and physiological studies. Previous research indicates that *AMIGO2* regulates cell survival, adhesion, migration, and angiogenesis, significantly impacting cell proliferation and tissue development [74]. In cancer biology, *AMIGO2* promotes tumor angiogenesis and glucose metabolism, suggesting that it may modulate vascularization and nutrient uptake in energy-demanding tissues [75].

Despite DRC pigs being widely recognized as lean pigs known for rapid growth, the lineage-specific expression and epigenetic regulation of *AMIGO2* identified in BF suggest a potential role for this gene in the regulation of adipose tissue development. Additional phenotypic measurements indicated that hybrids of DRC and LG exhibited significantly thicker backfat at 90–100 days compared to other hybrids (Appendix A), implying an earlier onset of adipocyte differentiation and backfat development in hybrids containing DRC lineage. Furthermore, the eQTLs of *AMIGO2* identified in this study overlapped with previously reported eQTLs in pig adipose tissue [59], both of which were located within lineage-specific H3K27ac peaks. It suggests that lineage-specific enhancers marked by H3K27ac may play a critical role in regulating *AMIGO2* expression in adipose tissue. The strong alignment observed among gene expression, phenotype, and epigenetic modifications indicates that *AMIGO2* could be a candidate gene affecting pig fat development, providing essential clues for understanding lineage-specific genetic regulatory mechanisms.

Importantly, our findings have direct implications for pig breeding and commercial production. By identifying lineage-specific regulatory variants and their downstream effects on gene expression and tissue development, this study provides molecular targets that can be leveraged in marker-assisted selection (MAS) and genomic selection (GS) programs. For instance, the strong association of *KIT* expression with both fat and muscle development highlights its potential as a dual-function marker for optimizing meat quality and yield traits. Similarly, the lineage-specific activation of *AMIGO2* in DRC hybrids suggests an opportunity to fine-tune adipose tissue development in lean-type breeds without compromising growth rate. The early onset of backfat accumulation in DRC×LG hybrids at 90–100 days, a critical window in commercial fattening stages, underscores the practical value of phase-informed regulatory elements for early trait prediction and selection. Furthermore, by integrating pedigree-based phasing and multi-omic data, our strategy provides a cost-effective and scalable framework applicable to other livestock species with structured breeding systems. These insights not only deepen our understanding of genetic regulation in hybrids but also translate into tangible benefits for enhancing production efficiency and trait optimization in modern pig breeding.

## 5. Conclusions

This study systematically analyzed phased genetic variants in crossbred pigs and explored their roles in gene expression and epigenetic modifications. The findings not only reveal how genetic variants affect lineage-specific gene expression but also provide potential genetic markers for molecular breeding, providing a more comprehensive theoretical framework for pig breeding and genomic research.

## Figures and Tables

**Figure 1 animals-15-01494-f001:**
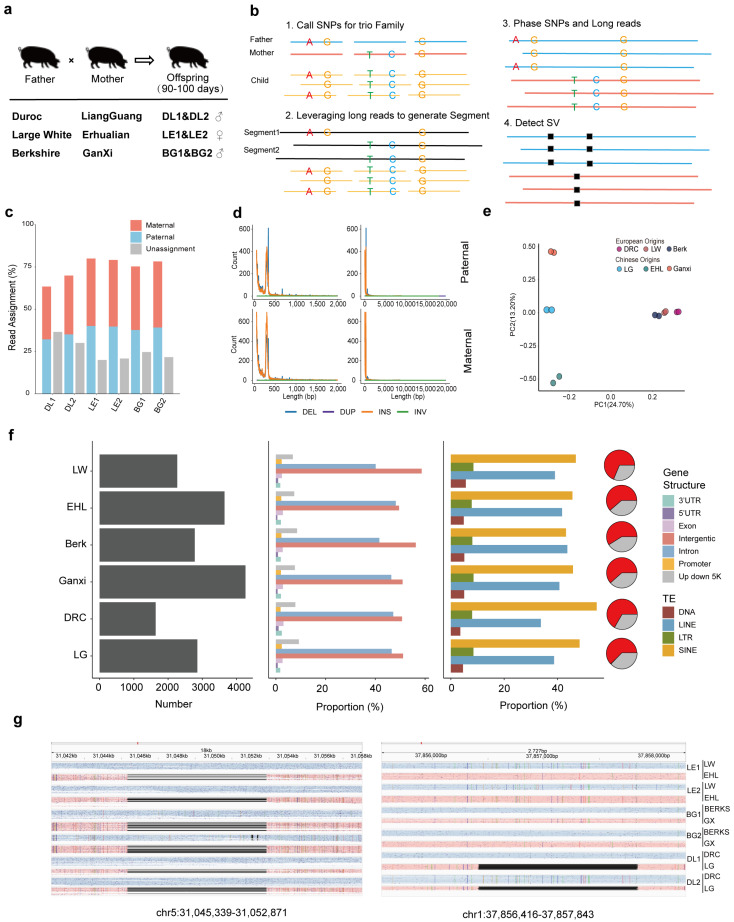
Identification and characteristics of genetic variations from different lineages. (**a**) Design of Central European distant hybrid family. (**b**) Strategy for identifying phased genetic variations (SNPs and SVs). (**c**) Phasing efficiency of ONT data for SV identification. (**d**) Length distribution of SVs on parental phases and proportion of different variant event annotations. (**e**) Clustering of SVs detected from different lineages. (**f**) Number of lineage-specific SVs detected in phases from different lineages (left), their proportions in different gene structures (middle), their proportions in different transposable elements (right), and pie plot showed the percentage of phased SVs contained within transposable elements in red and others in grey. (**g**) Identification of phased SVs longer than 1 kb with ONT data, maternal-phase-specific DEL (left) and LG-specific DEL (right).

**Figure 2 animals-15-01494-f002:**
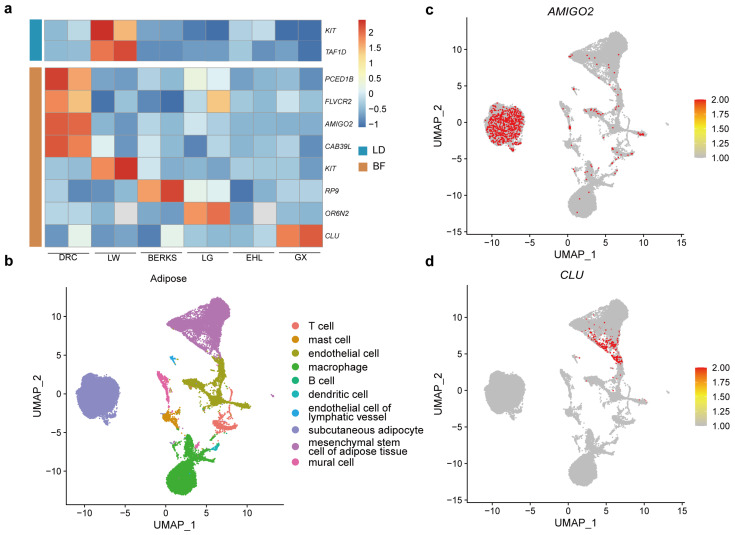
Lineage-specific gene expression and functional annotation. (**a**) Lineage-specific gene expression was found to be significantly associated with lineage-specific genetic variants. (**b**) UMAP plot depicting cellular composition of adipose based on scRNA-seq data. (**c**) Major cell type expression of *AMIGO2* in adipose cells. (**d**) Major cell type expression of *CLU* in adipose cells.

**Figure 3 animals-15-01494-f003:**
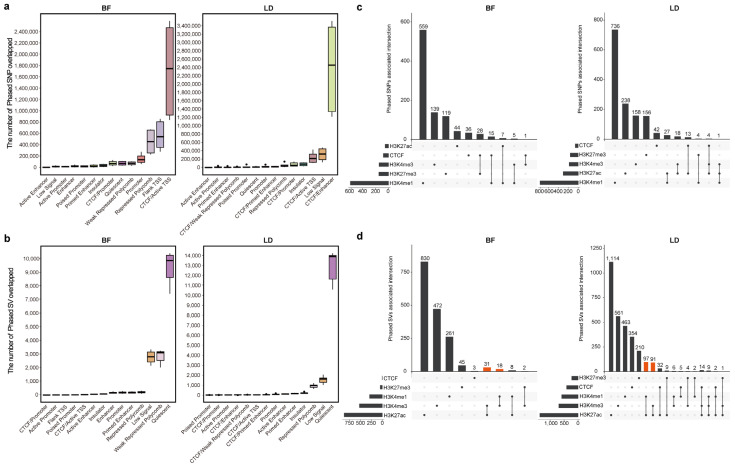
Distribution of genetic variants in different chromatin active regions and colocalization between phased genetic variants and phased histone modification and CTCF signals. (**a**) Distribution of SNPs in different chromatin states. (**b**) Distribution of SVs in different chromatin states. (**c**) Colocalization of phased SNPs with one or more phased histone modifications and CTCF signals (BF, left; LD, right). (**d**) Colocalization of phased SVs with one or more phased histone modifications and CTCF signals (BF, left; LD, right; Orange bars show the active chromatic indicator).

**Figure 4 animals-15-01494-f004:**
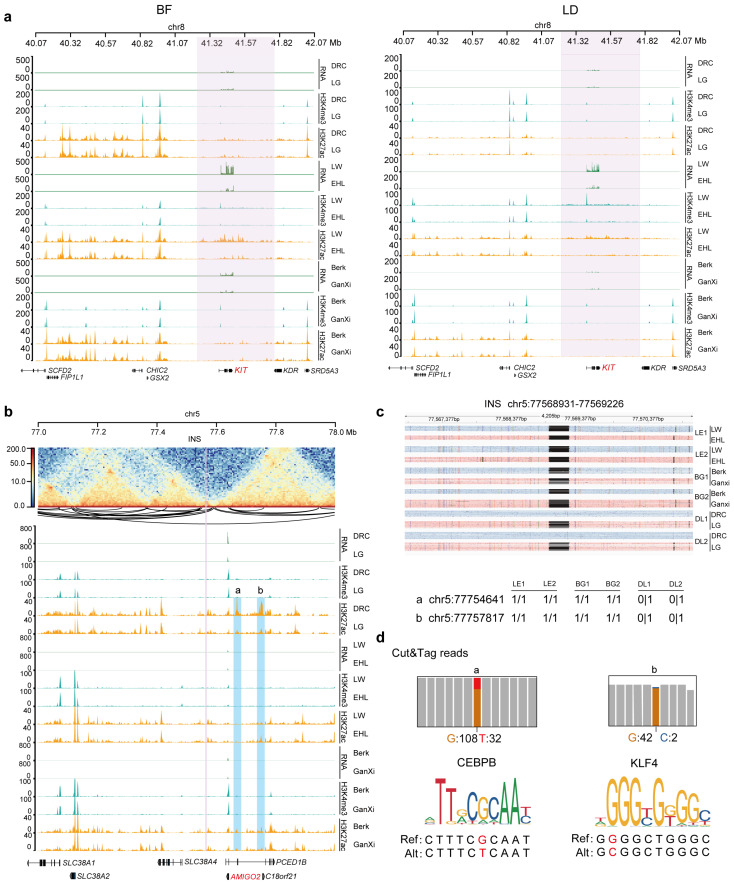
Lineage-specific SV influences lineage-specific expression mechanisms and prediction of molecular markers. (**a**) KIT gene expression levels from different lineages and associated H3K4me3 and H3K27ac signals. Pink highlights indicate lineage-specific SV regions from LW lineage. (BF: left; LD: right) (**b**) AMIGO2 gene expression levels in BF from different lineages, surrounding H3K4me3 and H3K27ac signals, and chromatin 3D interactions. Pink highlights INS significantly associated with expression and histone modification signals in phase from DRC lineage. Two lineage-specific H3ak27ac peaks that covered SNPs (*a* and *b*) highlighted in blue show consistent associations with same signals as INS. (**c**) Genetic variations related to histone modification signals in AMIGO2 gene and its upstream regions across different phases of lineages. (**d**) Upper panel shows H3K27ac signal coverage at SNPs *a* and *b* in different phases. Lower panel presents predicted core motifs of transcription factors.

**Table 1 animals-15-01494-t001:** Association of SNP and SV variants in different chromosomal regions with gene expression, epigenetic modifications, and tissues.

SNP Region (Clustered by Window of 1 Mb)	SV Region	Gene	H3K4me3	H3K27ac	H3K4me1	H3K27me3	CTCF	Tissue
chr5:77483571-78328698	(INS) chr5:77568931-77569226	*AMIGO2*	-	chr5:77754058-77755070 chr5:77757811-77758325	-	-	-	BF
chr7:135107363-135351353	-	*ENSSSCG00000036983*	-	-	-	-	chr7:135297819-135299247	BF
-	(DUP) chr8:41223208-41783661	*KIT*	chr8:41401921-41403769	chr8:41401817-41403400	-	-	-	BF
	(DUP) chr8:41223208-41783661	*KIT*	chr8:41401866-41403769	chr8:41402038-41403489	-	-	-	LD
chr12:19702952-20479694	-	*PSME3*	-	chr12:19713381-19714949	-	-	chr12:19713354-19714899	LD

**Table 2 animals-15-01494-t002:** Allele frequencies of SNPs and SVs in different chromosomal regions in large populations corresponding to different Chinese and European breeds, respectively.

Breeds	Variants	Ref_Allele_Freq (%)	Alt_Allele_Freq (%)
DRC	chr5:77568931-77569226	71.08	28.92
LW		20.81	79.19
LaiWu		13.64	86.36
EHL		8.02	91.98
DRC	chr5:77754641-77754642	90.17	9.83
LW		1.94	98.06
LaiWu		2.27	97.73
EHL		21.96	78.04
DRC	chr5:77757817-77757818	90.03	9.97
LW		0.00	100.00
LaiWu		2.27	97.73
EHL		13.08	86.92

## Data Availability

The data provided in this study can be downloaded from publicly accessible databases and partially requested from the corresponding author.

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
