# Peer review of "Parental Phasing Study Identified Lineage-Specific Variants Associated with Gene Expression and Epigenetic Modifications in European–Chinese Hybrid Pigs"

_animals, 2025, doi:10.3390/ani15101494_

Round 1

Reviewer 1 Report

Comments and Suggestions for Authors

The authors presented a concise and well-structured manuscript, with methodologies and results clearly defined and discussed in detail. This study is innovative in its approach to studying heterosis in crossbred pigs, and their analysis and integration of phased genetic and structural variants with gene expression and epigenetic modifications is sound. However, multiple errors in the text require attention, and the methods should include additional details on the sequencing technology used in the study. Below is a list of my recommended corrections.

What Illumina sequencing platform was used for RNA and DNA sequencing? Was the same one used for Hi-C data? Was the RNA and DNA sequencing also done as 150bp paired-end reads?

Gene symbols should be defined in the manuscript or a legend.

Figure references are inconsistent in format. Change “Figure” to “Fig.” for all references to figure 1 to align with the format used for the other figures.

Line 35: Change “Integrating of eQTL and epiQTL” to “Integration of eQTL and epiQTL”

Lines 74-76: “Genetic variants differences between parental phases may contribute to hybrid offspring’s unique trait advantages through multiple mechanisms.” Rephase sentence for clarity.

Line 99: Change was to were in “Mendelian errors was calculated…”.

Line 176: Define TAD, i.e. topologically associated domains.

Line 229: Remove be in “SVs were also be phased”.

Line 280: Remove “there are” in “Notably, there are colocalizations between phased …”.

Line 283: Rewrite the sentence “To further functional annotate the genes, we integrated single-cell transcriptomic data”, (e.g. Additional functional annotation was performed by integrating single-cell transcriptomic data …)

Lines 337-338: The sentence starting with  “As the differential distribution of SNPs and SVs across chromatin regions with distinct activities, …” should be rewritten for clarity.

Line 384: Change covered to covering in “motif prediction for peaks covered a and b …” and change “found that the SNP a located in” to “found that the SNP a was located in”.

Line 387: Change covered to covering in “peaks covered a and b”.

Line 388: Change “H3K27ac peak covered b and the promoter …” to “H3K27ac peak covering SNP b with the promoter …”.

Line 397: Change covered to covering in “within the H3K27ac peak covered a.”.

Lines 414-415: Change “INS associated with significantly associated with expression …” to “INS significantly associated with expression …”.

Lines 446-447: Rephrase sentence “Compared with conventional methods[58-60], eQTL mapping with phased gene expression could offer distinct advantages by substantially power improvement[61].”

Author Response

Reviewer1

Comments 0: The authors presented a concise and well-structured manuscript, with methodologies and results clearly defined and discussed in detail. This study is innovative in its approach to studying heterosis in crossbred pigs, and their analysis and integration of phased genetic and structural variants with gene expression and epigenetic modifications is sound. However, multiple errors in the text require attention, and the methods should include additional details on the sequencing technology used in the study. Below is a list of my recommended corrections.

Response 0: We sincerely appreciate the reviewer's recognition of our work and the positive feedback provided. We also apologize for the errors in the manuscript and the lack of detail in our methodology. We have carefully addressed the reviewer's suggestions and made the necessary revisions and additions to improve the manuscript.

Comments 1: What Illumina sequencing platform was used for RNA and DNA sequencing? Was the same one used for Hi-C data? Was the RNA and DNA sequencing also done as 150bp paired-end reads?

Response 1: In our study, all sequencing was performed using the 150 bp paired-end (PE150) strategy. The sequencing platforms used for different data types were as follows: Whole-genome sequencing (WGS), RNA sequencing (RNA-seq), and Hi-C libraries were sequenced on the Illumina HiSeq 4000 platform. CUT&Tag libraries were sequenced on the Illumina NovaSeq 6000 platform. We have added detailed descriptions regarding the sequencing platforms and strategies used for each data type in the revised Methods section, as shown below.

“Whole-genome short-read libraries were sequenced on the Illumina HiSeq 4000 platform (PE150), and long-read sequencing was performed on the ONT PromethION platform.” (Line 140-142)

“Total RNA was extracted and purified, mRNA was enriched using oligo(dT) beads and converted into cDNA. Libraries were constructed using the NEBNext® Ultra™ RNA Library Prep Kit for Illumina (NEB, USA). RNA-seq libraries were sequenced on the Illumina HiSeq 4000 platform using 150 bp paired-end reads” (Line 179-182)

“Cut&Tag were conducted on five hybrid offspring, with a total of 50 samples from backfat (BF) and longissimus dorsi (LD) muscle, to profile multiple histone modifications and CTCF binding. Libraries were constructed for each target and sequenced on the Illumina NovaSeq 6000 platform using 150 bp paired-end reads.” (Line 192-195)

Comments 2: Gene symbols should be defined in the manuscript or a legend.

Response 2: We appreciate your valuable suggestion. We have added the full gene names upon their first appearance in the Results sections. To maintain conciseness throughout the manuscript, the remaining gene symbols are presented in their abbreviated form.

“Notably, colocalizations between phased genetic variants and phased gene expressions were lineage-specific, including AMIGO2 (adhesion molecule with Ig like domain 2), KIT (KIT proto-onco, receptor tyrosine kinase), OR6N2 (olfactory receptor family 6 subfamily N member 2), FLVCR2 (FLVCR choline and putative heme transporter 2), CLU (clusterin), PCED1B (PC-esterase domain containing 1B), CAB39L (calcium binding protein 39 like) and RP9 (RP9 pre-mRNA splicing factor) in BF, and TAF1D (TATA-box binding protein associated factor) and KIT in LD (Fig. 2a).”(Line 330-336)

Comments 3: Figure references are inconsistent in format. Change “Figure” to “Fig.” for all references to figure 1 to align with the format used for the other figures.

Response 3: The revision has been completed.

Comments 4: Line 35: Change “Integrating of eQTL and epiQTL” to “Integration of eQTL and epiQTL”

Response 4: The revision has been completed.

Comments 5: Lines 74-76: “Genetic variants differences between parental phases may contribute to hybrid offspring’s unique trait advantages through multiple mechanisms.” Rephase sentence for clarity.

Response 5: Thank you for the suggestion. The sentence has been rephrased for clarity as follows: “Differences in genetic variants between parental phases may contribute to the unique trait advantages of hybrid offspring through various mechanisms.” (Line 87-89)

Comments 6: Line 99: Change was to were in “Mendelian errors was calculated…”.

Response 6: The revision has been completed.

Comments 7: Line 176: Define TAD, i.e. topologically associated domains.

Response 7: Thank you for the suggestion. TAD has been defined as topologically associated domains.

Comments 8: Line 229: Remove be in “SVs were also be phased”.

Response 8: The revision has been completed

Comments 9: Line 280: Remove “there are” in “Notably, there are colocalizations between phased …”.

Response 9: The revision has been completed

Comments 10: Line 283: Rewrite the sentence “To further functional annotate the genes, we integrated single-cell transcriptomic data”, (e.g. Additional functional annotation was performed by integrating single-cell transcriptomic data …)

Response 10: Thank you for the suggestion. The sentence has been rewritten as follows for clarity and flow:

“Additional functional annotation was performed by integrating single-cell transcriptomic data from human adipocytes[1] and skeletal muscle[2] along with GWAS datasets (https://pigbiobank, https://www.ebi.ac.uk/gwas/home).” (Line 336-337)

Comments 11: Lines 337-338: The sentence starting with “As the differential distribution of SNPs and SVs across chromatin regions with distinct activities, …” should be rewritten for clarity.

Response 11: We apologize for the lack of clarity in the original sentence. It has been revised as follows: “We then performed co-localization analysis using phased genomic variants (SNPs and SVs) with phased histone modifications and CTCF peaks, respectively. For SNPs, we directly examined whether they overlapped with these peaks. For SVs, since most are located in inactive chromatin regions, we checked whether phased SVs were within ±1 Mb of these peaks.” (Line 391-395)

Comments 12: Line 384: Change covered to covering in “motif prediction for peaks covered a and b …” and change “found that the SNP a located in” to “found that the SNP a was located in”.

Response 12: The revision has been completed

Comments 13: Line 387: Change covered to covering in “peaks covered a and b”.

Response 13: The revision has been completed

Comments 14: Line 388: Change “H3K27ac peak covered b and the promoter …” to “H3K27ac peak covering SNP b with the promoter …”.

Response 14: The revision has been completed

Comments 15: Line 397: Change covered to covering in “within the H3K27ac peak covered a.”.

Response 15: The revision has been completed

Comments 16: Lines 414-415: Change “INS associated with significantly associated with expression …” to “INS significantly associated with expression …”.

Response 16: The revision has been completed

Comments 17: Lines 446-447: Rephrase sentence “Compared with conventional methods[58-60], eQTL mapping with phased gene expression could offer distinct advantages by substantially power improvement[61].”

Response 17: Thank you for the suggestion. The sentence has been rephrased: “Compared with conventional methods[3-5], eQTL mapping using phased gene expression offers significant advantages, primarily by greatly enhancing statistical power [6].” (Line 505-506)

Reviewer 2 Report

Comments and Suggestions for Authors

The statement chinese breeds are superior is way to general and is fundamentally illogical given the hundreds of chinese breeds of pigs - all with their own qualities.

The breeds of pig selected are also generally obscure and need defining

Why were the pigs slaughtered so young at 90-100 days this is not commercially viable.

Ensure spelling is correct in document.

A lot of nice 'science' and graphs but must make the work commercially relevant

FDR needs more defining for the general reader

Author Response

Reviewer2

Comments 1: The statement chinese breeds are superior is way to general and is fundamentally illogical given the hundreds of chinese breeds of pigs - all with their own qualities.

Response 1: Thank you for your insightful comment. We agree that the original statement was overly generalized. In the revised version, we have modified the sentence to more accurately reflect the diversity among Chinese pig breeds.

“For instance, several representative Chinese indigenous pig breeds, such as Erhualian and Meishan, are well known for their high reproductive performance and enhanced fat deposition, whereas European commercial pigs, including breeds like Landrace and Duroc, are characterized by faster growth rates and higher lean meat yield[7-9].” (Line 89-93)

Comments 2: Need to define acronyms. BF and LD???? The average reader may not understand even if put into abstract.

Response 2: Thank you for your suggestion. We have defined the acronyms ‘BF’ (backfat) and ‘LD’ (longissimus dorsi) upon their first appearance in the abstract, methods, and results sections to ensure clarity for all readers. For the sake of conciseness, the abbreviations are used throughout the remainder of the manuscript. It also be seen below:

“We collected semen from the father, ear tissue from the mother, and backfat (BF) and longissimus dorsi (LD) muscle samples from hybrid offspring…” (Line 127-128)

Comments 3: The breeds of pig selected are also generally obscure and need defining

Response 3: We thank you for your suggestion and have added a brief description of the breeds in the Materials section as follows:

“In this study, three hybrid families were generated by crossing representative European commercial breeds with Chinese indigenous breeds. The paternal lines included Duroc Large White and Berkshire, which are commonly used in commercial pig breeding due to their well-established performance in growth and carcass traits. The maternal lines consisted of three geographically distinct Chinese native breeds. Erhualian pigs, renowned for their high reproductive capacity, originate from Jiangsu Province in eastern China. GanXi pigs, characterized by their typical 'two-end black' appearance, are native to western Jiangxi Province. LiangGuang pigs, distinguished by their small ears and piebald coat, are distributed across Guangdong and Guangxi Provinces in southern China.” (Line 118-126)

Comments 4: Why were the pigs slaughtered so young at 90-100 days this is not commercially viable.

Response 4: Thank you for your concern. Our study primarily focused on gene expression during the rapid growth phase, as we aimed to explore lineage-specific developmental mechanisms at this stage. This period was chosen to gain deeper insights into the genetic and epigenetic processes governing growth and development. Although the slaughter age may not be commercially viable, it was essential for understanding the early developmental stages critical to our research objectives.

Q5: Ensure spelling is correct in document.

Response: We apologize for the spelling errors, and they have been corrected.

Comments 5: A lot of nice 'science' and graphs but must make the work commercially relevant

Response 5: We thank the reviewer for pointing this out. To address this comment, we have revised the Discussion section to emphasize the potential commercial relevance of our findings.

“Importantly, our findings have direct implications for pig breeding and commercial production. By identifying lineage-specific regulatory variants and their downstream effects on gene expression and tissue development, this study provides molecular targets that can be leveraged in marker-assisted selection (MAS) and genomic selection (GS) programs. For instance, the strong association of KIT expression with both fat and muscle development highlights its potential as a dual-function marker for optimizing meat quality and yield traits. Similarly, the lineage-specific activation of AMIGO2 in DRC hybrids suggests an opportunity to fine-tune adipose tissue development in lean-type breeds without compromising growth rate. The early onset of backfat accumulation in DRC×LG hybrids at 90–100 days, a critical window in commercial fattening stages, underscores the practical value of phase-informed regulatory elements for early trait prediction and selection. Furthermore, by integrating pedigree-based phasing and multi-omic data, our strategy provides a cost-effective and scalable framework applicable to other livestock species with structured breeding systems. These insights not only deepen our understanding of genetic regulation in hybrids but also translate into tangible benefits for enhancing production efficiency and trait optimization in modern pig breeding.” (Line 562-577)

Comments 6: FDR needs more defining for the general reader

Response 6: Thank you for your comment. To improve clarity for general readers, we have added a brief definition of the false discovery rate (FDR) in the Methods section of the manuscript. We also indicated “see Methods” upon its first mention in the Results section.

“Significant associations were defined as those with a FDR less than 0.05, a statistical method that controls for multiple testing by limiting the expected proportion of false positives[10].” (Line 239-241)

Reviewer 3 Report

Comments and Suggestions for Authors

Detailed comments

Title of the article

The title of the article reflects the subject of the research.

Abstract

The abstract should be thoroughly revised. The text does not refer directly to the results. The most important results with statistics are not provided. There is no conclusion summarizing the research.

Introduction

On the basis of what scientific premises did they choose histone modifications (H3K4me3, H3K27ac, H3K4me1 and H3K27me3) for the research? Please distinguish the research hypothesis and the aim of the research.

Research methodology

In the methodology, it is not stated on how many pigs in each experimental group the molecular analyses were performed. The analytical methods used are appropriate for this type of research.

Results

The results were collected in 4 graphs and 2 tables with a description.

Discussion

The discussion requires improvement, the obtained results concerning the KIT and AMIGO2 genes KIT and AMIGO2 genes should be discussed in more detail with the scientific literature in this field.

Final note

The work requires supplementation and improvement.

Author Response

Reviewer3

Comments 1: The abstract should be thoroughly revised. The text does not refer directly to the results. The most important results with statistics are not provided. There is no conclusion summarizing the research.

Response 1: We thank the reviewer for the constructive suggestions. We revised a sentence “Understanding how hybrids integrate lineage-specific regulatory variants at the haplotype level is crucial for elucidating the genetic basis of heterosis in livestock” (Line 28-29) to better emphasize the scientific objective and significance of our study.

In addition, we have added statistical significance values to key results. Specifically, we now state: “By colocalization analysis of phased genetic variants with phased gene expression levels and with phased epigenetic modifications, we identified 18,670 expression quantitative trait loci (eQTL)(FDR<0.05) and 8,652 epigenetic modification quantitative trait loci (epiQTL)(FDR<0.05).” (Line 36-37)

“For example, we identified a Large White lineage-specific duplication (DUP) encompassing the KIT gene that was significantly associated with its promoter activity (FDR= 7.83 × 10⁻⁴) and expression levels (FDR= 9.03 × 10⁻⁴). Additionally, we found that a Duroc lineage-specific SNP located upstream of AMIGO2 was significantly associated with a Duroc-specific H3K27ac peak (FDR = 0.035), and also showed a significant association with AMIGO2 expression levels (FDR = 5.12 × 10⁻⁴).” (Line 39-44)

Finally, we added a concluding sentence to the abstract to summarize the overall findings and their implications: “These findings underscore the importance of phased regulatory variants in shaping lineage-specific transcriptional programs, and highlight how haplotype-resolved integration of eQTL and epigenetic signals can reveal the mechanistic underpinnings of hybrid regulatory architecture. Our results offer insights for molecular marker development in precision pig breeding.” (Line 44-48)

Comments 2: On the basis of what scientific premises did they choose histone modifications (H3K4me3, H3K27ac, H3K4me1 and H3K27me3) for the research? Please distinguish the research hypothesis and the aim of the research.

Response 2: We sincerely thank the reviewer for the insightful comments and constructive suggestions.

The scientific premise underlying the selection of histone modifications (H3K4me3, H3K27ac, H3K4me1, and H3K27me3) is that these marks represent distinct chromatin states associated with gene regulatory and chromatin activity. Specifically, H3K4me3 is enriched at active promoters, H3K27ac and H3K4me1 are hallmarks of active and primed enhancers, respectively, while H3K27me3 is associated with Polycomb-mediated gene repression. By integrating both activating and repressive marks, we aimed to capture a comprehensive view of the epigenetic regulation landscape at the haplotype level.

The research hypothesis is that numerous studies have found that histone modifications exhibit haplotype-specific patterns in hybrid organisms, reflecting parental differences in epigenetic regulation, which contribute to allele-specific gene expression and the variation of hybrid traits.

The aim of the research was to generate haplotype-resolved chromatin profiles for key histone modifications in hybrid pigs and to identify lineage-specific regulatory elements that may underlie phenotypic variation, thereby elucidating the epigenetic mechanisms driving in hybrid vigor.

We have also incorporated the relevant research background and the scientific premise into the Introduction section, as shown below:

“In agricultural breeding programs for plants and animals, the superior traits of hybrid offspring are often associated with gene expression differences between paternal and maternal lineages. Functional annotation and mechanistic analysis of regulatory elements at the haplotype level allow for more precise identification of parental alleles that contribute to specific traits, such as disease resistance, productivity, or product quality. Li et al. conducted a comprehensive haplotype-resolved annotation of H3K27me3 in the hybrid rice line[11]. In studies of major livestock (pigs [12]) and poultry (chickens [13]), researchers similarly used hybrid offspring to distinguish between parental genomic contributions at the haplotype level. They examined multiple histone modifications, primarily H3K4me3 and H3K27ac, and observed signal differences between paternal and maternal haplotypes. These findings underscore the importance of integrating diverse histone modification types to characterize haplotype-specific regulatory mechanisms.” (Line 99-110)

Comments 3: In the methodology, it is not stated on how many pigs in each experimental group the molecular analyses were performed. The analytical methods used are appropriate for this type of research.

Response 3: We thank the reviewer for the valuable comment. We apologize for the omission and have now included detailed information regarding the number of pigs used for each molecular analysis in the revised Methods section. Specifically, 16 pigs were used for short-read WGS (both parents and offsprings). For offsprings, 6 pigs were long-read WGS, 6 pigs for RNA-seq and 5 pigs for CUT&Tag. Additionally,1 adult pig was used for Hi-C analyses, which was mentioned in the original manuscript. Others could be seen below:

“A total of 16 pigs, including both parents and offspring, were used for short-read whole-genome sequencing (WGS), while long-read WGS was performed on 6 offspring individuals.” (Line 137-139)

“RNA-seq was performed on six hybrid offspring, with both backfat (BF) and longissimus dorsi (LD) muscle collected from each individual, resulting in a total of twelve samples.” (Line 177-179)

“Cut&Tag were conducted on five hybrid offspring, with a total of 50 samples from backfat (BF) and longissimus dorsi (LD) muscle, to profile multiple histone modifications and CTCF binding.” (Line 192-194)

“”

Comments 4: The discussion requires improvement, the obtained results concerning the KIT and AMIGO2 genes KIT and AMIGO2 genes should be discussed in more detail with the scientific literature in this field.

Response 4: Thank you for the insightful comments. We have expanded the discussion to more comprehensively address the functional relevance of KIT and AMIGO2 in adipose and muscle development, which are key biological processes central to the focus of our study.

“Importantly, KIT has also been implicated in muscle satellite cell function and adipose tissue remodeling in large-animal models. For instance, Ma et al reported that KIT is the key genes of subcutaneous fat in cattle[14], suggesting a conserved role in livestock fat deposition. In pigs, CNVs and duplications in the KIT locus have been associated not only with coat color variation but also with differences in body composition traits, including backfat thickness and carcass quality[15]. Recent analyses further suggest that KIT may regulate early adipocyte differentiation through crosstalk with Wnt and PI3K-Akt pathways, which are key to fat tissue development[16,17]. These evidences suggest that beyond its well-known role in coat color formation[18,19], KIT may also play a critical biological role in fat deposition, muscle cell differentiation, and immune function in pigs.”(Line 529-536)

“Although research on AMIGO2 in agricultural animals remains limited, it is considered to be highly associated with cellular development and metabolism in medical and physiological studies. Previous research indicates that AMIGO2 regulates cell survival, adhesion, migration, and angiogenesis, significantly impacting cell proliferation and tissue development[20]. In cancer biology, AMIGO2 promotes tumor angiogenesis and glucose metabolism, suggesting that it may modulate vascularization and nutrient uptake in energy-demanding tissues[21].” (Line 541-547)

Round 2

Reviewer 2 Report

Comments and Suggestions for Authors

Nice interesting article